# Increased Phosphatase of Regenerating Liver-1 by Placental Stem Cells Promotes Hepatic Regeneration in a Bile-Duct-Ligated Rat Model

**DOI:** 10.3390/cells10102530

**Published:** 2021-09-24

**Authors:** Jong Ho Choi, Sohae Park, Gi Dae Kim, Jae Yeon Kim, Ji Hye Jun, Si Hyun Bae, Soon Koo Baik, Seong-Gyu Hwang, Gi Jin Kim

**Affiliations:** 1Department of Oral Pathology, College of Dentistry, Gangneung-Wonju National University, Gangneung-si 25457, Korea; jhchoi@gwnu.ac.kr; 2Department of Biomedical Science, CHA University, Seongnam-si 13488, Korea; sohae11@snu.ac.kr (S.P.); janejaeyeon92@gmail.com (J.Y.K.); jihyejun1015@gmail.com (J.H.J.); 3Research Institute of Placental Science, CHA University, Seongnam-si 13488, Korea; 4Department of Food and Nutrition, Kyungnam University, Changwon-si 51767, Korea; gidaekim@kyungnam.ac.kr; 5Department of Internal Medicine, Catholic University Medical College, Seoul 03312, Korea; baesh@catholic.ac.kr; 6Department of Internal Medicine, Yonsei University Wonju College of Medicine, Wonju 26426, Korea; baiksk@yonsei.ac.kr; 7CHA Bundang Medical Center, Department of Internal Medicine, Division of Gastroenterology, CHA University School of Medicine, Seongnam-si 13496, Korea; sghwang@cha.ac.kr

**Keywords:** chorionic-plate-derived mesenchymal stem cells, phosphatase of regenerating liver-1, liver regeneration, bile duct ligation

## Abstract

Phosphatase of regenerating liver-1 (PRL-1) controls various cellular processes and liver regeneration. However, the roles of PRL-1 in liver regeneration induced by chorionic-plate-derived mesenchymal stem cells (CP-MSCs) transplantation remain unknown. Here, we found that increased PRL-1 expression by CP-MSC transplantation enhanced liver regeneration in a bile duct ligation (BDL) rat model by promoting the migration and proliferation of hepatocytes. Engrafted CP-MSCs promoted liver function via enhanced hepatocyte proliferation through increased PRL-1 expression in vivo and in vitro. Moreover, higher increased expression of PRL-1 regulated CP-MSC migration into BDL-injured rat liver through enhancement of migration-related signals by increasing Rho family proteins. The dual effects of PRL-1 on proliferation of hepatocytes and migration of CP-MSCs were substantially reduced when PRL-1 was silenced with siRNA-PRL-1 treatment. These findings suggest that PRL-1 may serve as a multifunctional enhancer for therapeutic applications of CP-MSC transplantation.

## 1. Introduction

Despite the liver’s strong regenerative capacity, liver fibrosis and hepatic failure are often caused by various agents, such as inflammation, parasites, metabolic toxins, or drugs, as well as vascular or congenital defects [1,2,3]. Liver transplantation is well known as an optimal therapy for decompensated cirrhotic liver and acute liver failure [4], but severe side effects, such as immune responses, can lead to allograft failure [5,6,7,8]. Additionally, it could induce several complications of biliary nature [9]. Therefore, alternative strategies are required for the treatment of liver disease.

The phosphatase of regenerating liver (PRL) family is a group of protein tyrosine phosphatases (PTPs) that consists of three closely related members, PRL-1, PRL-2, and PRL-3, which share 76% to 87% amino acid identity [10]. Specially, PRL-1 was first discovered and identified as a critical factor in mitogen-stimulated cells during liver regeneration [11]. Following the first study, initial studies about PRL-1 have reported it to have oncogenic roles in cancer. For example, overexpression of PRL-1 was observed in several kinds of cancers and correlated with poor patient prognosis, including in gastric, intrahepatic, and prostate cancer [12,13,14]. Recently, most PRL-1 studies have focused on the role of PRL-1 on cell migration in cancer studies. Zeng Q et al. showed that PRL-1-expressing cells induce metastatic tumor formation in mice [15]. Mechanistically, PRL-1 is related to several major signaling pathways of cell migration, such as Src, Rho family GTPases, and adhesion molecules and enzymes [16]. Inhibition of PRL-1 expression reduces migration and invasion through c-Src and Rho family GTPases, including Rac1 and cdc42, in cancer cells [17]. However, overexpression of PRL-1 induces cell migration ability by regulating the E-cadherin and Rho family GTPases, as well as causing the overexpression of PRL-1, which was observed in metastatic tissue compared to non-metastatic tissue, in consistent agreement with the role of PRL-1 in the regulation of cell migration. Another function of PRL-1 is to promote cell migration and invasion by regulating the matrix metalloproteinases (MMP). MMPs are extracellular secreted proteins, which are a key marker in cell migration as well as tumor metastasis [18]. Furthermore, overexpressed PRL-1 expression promotes the increased cell migration and invasion ability via stimulation of MMP-2 and MMP-9 activity [19], which is in agreement with a role of PRL-1 in the regulation of cell migration through gene expression, involved with cell migration.

The improvement in engraftments via the migration of stem cells into target tissues is one of the critical factors for maximizing the therapeutic effect of stem cells transplantation in regenerative medicine [20]. Specifically, the ability of mesenchymal stem cells (MSCs) to migrate to ischemic or damaged organs determines the therapeutic efficacy of MSCs, which is called “homing” [21]. Chemokines and cytokines released from damaged tissues provide cues for structural change in MSCs, leading to MSCs’ migration into ischemic or damaged organs, including the expression of a variety of adhesion molecules on the surface of MSCs [22,23]. The Rho family GTPases involved in cell migration drive the cytoskeletal remodeling of cells, which control the cell shape and mobility [24]. Activated Rho signaling induces a rearrangement of filamentous actin and myosin filaments, resulting in the activation of chemokine receptors [25]. Recently, Tan et al. reported that Rho family signaling induced by the C-X-C chemokine receptor type 4 (CXCR4) receptor is involved in stem cell migration, but mechanisms of MSCs homing induced by the microenvironment and its precise correlation between MSCs homing and repair of damaged liver remain unclear [26].

We previously reported that chorionic-plate-derived mesenchymal stem cells (CP-MSCs) isolated from normal-term placentas had potential anti-fibrosis, anti-inflammation, and hepatic regeneration effects via autophagic mechanisms in a carbon-tetrachloride (CCl_4_)-injured rat model [27,28]. Additionally, the hepatic proliferation effect of CP-MSC was demonstrated in hepatocytes damaged by lithocholic acid (LCA) in vitro [29]. The regulatory effects of PRL-1 on cell migration and potential therapeutic effects of CP-MSCs on hepatic regeneration may be correlated with the expression of PRL-1 and the Rho family signaling-mediated process. Therefore, we investigated PRL-1 expression in the bile duct ligation (BDL) rat model transplanted with CP-MSCs and the effects of PRL-1 on hepatic regeneration and migration of CP-MSCs on the therapeutic effects of CP-MSC transplantation, as well as in human cirrhotic liver. Additionally, we confirmed the expression and effect of the PRL-1 in vitro system using primary rat hepatocytes co-cultivated with CP-MSCs. Finally, their roles in hepatocytes and CP-MSCs were evaluated by siRNA-PRL-1 treatment and a co-cultivation system.

## 2. Materials and Methods

### 2.1. Human Liver Specimens

Human liver tissue from normal or cirrhotic patients was obtained under protocols approved by the Catholic University Medical College, Seoul, Korea. All participants provided written informed consent prior to sample collection. Liver tissue was collected from men or women who had no fibrosis (*n* = 5), minimal fibrosis (*n* = 5), mild fibrosis (*n* = 5), moderate fibrosis (*n* = 5), or severe cirrhosis (*n* = 5). This study was performed prospectively with approval from the Institutional Review Board (PC19OESI0015). All the tissue samples were collected, immediately snap-frozen in liquid nitrogen, and stored at −80 °C until RNA was extracted.

### 2.2. Generation of Animal Models

All animals were maintained in an air-conditioned animal house under specific pathogen-free conditions and starved of chow and water for 12 h before surgery and transplantation. For generation of the severe cirrhotic liver rat model, seven-week-old Sprague Dawley rats (Orient Bio Inc., Seongnam, Korea) were used to induce the common bile duct ligation (BDL) model. Briefly, the common bile duct was sectioned between ligatures to inhibit the extrahepatic biliary route for 10 days. After 10 days, CP-MSCs and WI-38 cells (2 × 10^6^ cells) were stained using a PKH67 Fluorescent Cell Linker Kit (0.5–10 µmol/L, Sigma-Aldrich, St. Louis, MO, USA), which stains cell membranes and is distributed equally to the daughter cells during division [30,31], and is injected into the tail vein (CP-MSCs; *n* = 36, WI-38; *n* = 41). PKH67+ cells (e.g., CP-MSCs and WI-38 cells) were analyzed using a fluorescence-activated cell sorter (FACS, BD Biosciences, San Jose, CA, USA). Non-transplanted (*n* = 20) rats were maintained as sham controls. Then, the liver tissues were collected at 1, 2, 3, and 5 weeks from transplanted and non-transplanted rats. A CCl_4_-injured rat model (*n* = 19) was used for a comparison group to assess PRL-1 expression in the BDL model. For the generation of the chemically induced cirrhotic liver rat model, the CCl_4_-injured rat model was used as previously described [27]. All animal experimental processes used protocols consistent with the Institutional Review Board of CHA General Hospital, Seongnam, Korea (IACUC-130009).

### 2.3. Serological Analysis

The collected human serum was examined for aspartate aminotransferase (AST), alanine aminotransferase (ALT), total bilirubin (TB), gamma glutamyl transpeptidase (γGT), prothrombin time (PT), and platelet count (PLT) by Gangnam CHA hospital (Seoul, Korea). The experiment was performed in triplicate.

### 2.4. Immunohistochemical Staining

Specimens of human and rat were fixed, embedded, and sectioned into slides (4 µm thick). Sections were deparaffinized with xylene and rehydrated and blocked with 3% H_2_O_2_ to block endogenous expression. After antigen epitope repairing, sections were blocked with 5% normal goat serum for 20 min, the slides were incubated overnight at 4 °C to the mouse anti-PRL1 (Abcam, Cambridge, UK). Sections were then incubated with biotinylated goat anti-mouse IgG (Santa Cruz, Dallas, TX, USA) for 1 h. Samples were stained with 3,3’-diaminobenzidine (VECTOR Laboratories. Burlingame, CA, USA), followed by counterstaining with hematoxylin. Sections were imaged under a light microscope (Zeiss, Oberkochen, Germany).

### 2.5. Cultivation of CP-MSCs and Primary Hepatocytes

Placentas were obtained from women who delivered at term (38 ± 2 gestational weeks). All participants provided written informed consent prior to the collection of placentas. The collection and use of placentas were approved by the Institutional Review Board of the Gangnam CHA General Hospital, Seoul, Korea (07–18). CP-MSCs were harvested as previously described [27]. Briefly, CP-MSCs were harvested from the inner side of the chorion-amniotic plate of the placenta by treatment with 0.5% collagenase IV (Sigma-Aldrich) for 30 min at 37 °C. The cells harvested from the placenta were cultured in Ham’s F-12/Dulbecco’s modified Eagle’s medium (Invitrogen, Carlsbad, CA, USA) with 1% P/S, (Invitrogen), 10% fetal bovine serum (FBS, Invitrogen), 1 µg/mL heparin (Sigma-Aldrich), and 25 ng/mL FGF-4 (Peprotech, Inc., Rocky Hill, NJ, USA). The WI-38 cells (ATCC, Manassas, VA, USA) were cultured with alpha-MEM (Invitrogen) supplemented with 1% P/S and 10% FBS. Isolated primary rat hepatocytes were cultured with William’s E medium (Lonza, Basel, Switzerland) supplemented with 10% FBS and 2% P/S at 37 °C.

### 2.6. In Vitro Co-Culture Experiments

Rat primary hepatocytes (5 × 10^5^ cells/mL) were seeded on 6-well culture plates (BD Falcon, NJ, USA). After 3 days of stabilization, the primary hepatocytes were incubated with 100 µM of LCA (Sigma-Aldrich) for 12 h and washed with 1× Dulbecco’s phosphate-buffered saline (DPBS). CP-MSCs or WI-38 cells (1 × 10^5^ cells/mL) were seeded with culture medium onto transwell inserts (BD Falcon) for 24 h at 37 °C. Additionally, to confirm the effect of PRL-1, 50 nM of siRNA-PRL-1 (Invitrogen) with lipofectamine (Invitrogen) was treated for 24 h in primary hepatocytes, comparing the mock control with a scrambled siRNA control.

### 2.7. Proliferation Assay Using BrdU Staining

For detection of proliferation of hepatocytes in injured livers after BDL, BrdU (50 mg/kg body weight; Sigma-Aldrich) was subcutaneously injected 2 h before sacrifice. BrdU-containing cells were detected by flow cytometry using a monoclonal mouse anti-BrdU antibody (Roche, Pleasanton, CA, USA). The percentage of proliferating cells was determined with a FACScan flow cytometer (BD Biosciences) and quantified with CellQuest software (BD Biosciences). For analysis of proliferating rat primary hepatocyte co-cultured with CP-MSCs and WI-38 cells, 10 µM BrdU was added at 37 °C for 2 h before harvest. BrdU-containing hepatocytes were determined by immunofluorescence analysis. BrdU-positive cells were counted and quantified with image processing software (ImageJ, NIH, Bethesda, MD, USA). The experiments were performed in triplicate.

### 2.8. Western Blot Analysis

Liver tissues from human and rat or primary hepatocytes co-cultured with CP-MSCs or WI-38 cells were homogenized and lysed in protein lysis buffer (Sigma) containing a protease inhibitor cocktail (Roche) and phosphatase inhibitor cocktail II (A.G Scientific, San Diego, CA, USA). Equal amounts of protein from liver samples were pooled from the samples. The protein lysates were loaded onto 10–15% sodium dodecyl sulfate polyacrylamide gels, and the separated proteins were transferred to PVDF membranes. Then, the PVDF membranes were blocked with 8% bovine serum albumin (BSA, Sigma) in 1× TBS (Sigma) for 1 h at room temperature and incubated overnight at 4 °C with primary antibodies. The primary antibodies were as follows: mouse anti-PRL1 (1:1000, Abcam); mouse anti-CXCR4 (1:1000, Abcam); mouse anti-albumin (1:2000, Santa Cruz); rabbit anti-Rho A (1:1000, Cell Signaling, Denvers, MA, USA); rabbit anti-ROCK1 (1:1000, Cell Signaling); and rabbit anti-GAPDH (1:3000, Young In Frontier Co., Ltd., Seoul, Korea). The membranes were washed with 1× TBS (Sigma) and incubated with a secondary antibody (horseradish peroxidase-conjugated anti-mouse IgG (1:10,000, Bio-Rad Laboratories, Hercules, CA, USA) or anti-rabbit IgG (1:5000, Bio-Rad Laboratories) for 1 h at room temperature. After the membranes were washed, bands were detected using enhanced chemiluminescence reagents (Amersham Biosciences, Pittsburgh, PA, USA). All reactions were performed in duplicate.

### 2.9. Enzyme-Linked Immunosorbant Assay (ELISA)

PRL-1 expression was quantified using a protein tyrosine phosphatase type IVA 1 (PTP4A1) ELISA kit (CUSABIO, Baltimore, MD, USA) per the manufacturer’s instructions. Briefly, human liver tissues were homogenized and lysed in 1× PBS (Abel-Bio, Seoul, Korea). Equal amounts of protein from individual humans were pooled, and 100 µL of the pooled sample was incubated for 2 h at 37 °C. After incubation, 100 µL of biotin–antibody (1×) was added and incubated for 1 h at 37 °C. The wells were washed twice with 200 µL of wash buffer for 2 min and incubated with 100µL of HRP–avidin (1×) for 1 h at 37 °C. Human PRL-1 expression was measured using a micro-plate reader (BioTek, Winooski, VT, USA) at 540 nm within 5 min. To analyze the expression of MMP-9 and MMP-2, we used the MMPs Quantikine ELISA (R&D Systems, Minneapolis, MN, USA) per the manufacturer’s instructions. Briefly, 100 µL of supernatant harvested from experimental transwells was added, mixed, and incubated at room temperature for 2 h. After incubation, the wells were washed three times with wash buffer, and 200 µL of MMP-9 and MMP-2 conjugate and substrate buffer were added to each well and incubated at room temperature for 2 h. Expression of MMP-9 and MMP-2 was measured using a micro-plate reader (BioTek) at 450 nm. All reactions were performed in triplicate. The data are expressed as the mean ± standard error (S.E.) of triplicate experiments.

### 2.10. Invasion of CP-MSCs Using a Transwell Insert System

To analyze the migration of CP-MSCs and WI-38 cells, we seeded CP-MSCs and WI-38 (3 × 10^4^) onto the upper inserts of a transwell insert system with or without siRNA-PRL1 (8 µm pore size; BD Falcon) after treatment with LCA in rat primary hepatocytes. CP-MSCs and WI-38 cells that infiltrated the lower insert were fixed with 100% methanol for 30 min and then stained with Mayer’s hematoxylin (Dako, Carpinteria, CA, USA) for 8 min. The number of stained cells was randomly counted in ten non-overlapping fields on the membranes at a magnification of 200×. The experiments were performed in triplicate.

### 2.11. ICG Clearance Test

Indocyanine green (ICG; Pulsion Medical Systems, Feldkirchen, Deutschland) was dissolved in sterile water to a final concentration of 5 mg/mL. ICG was injected into the tail vein in the BDL model and normal rat groups (0.5 mg/kg). Blood samples were obtained 15 min after ICG injection. Plasma samples (150 µL) were diluted with 750 µL of 1% bovine serum albumin and measured spectrophotometrically at 805 nm (Uvikon 850; Kontron instruments, Augsburg, Germany).

### 2.12. Gelatin Zymography

To analyze MMP-2 and MMP-9 activity in CP-MSCs and WI-38 cells, the supernatant was harvested in an invasion assay as described above. Each 20 µL of supernatant was separated in a 12% SDS–PAGE gel containing 1 mg/mLgelatin (Sigma). The gels were washed for 30 min using 1× renaturation buffer (Bio-Rad), rinsed, and incubated in 1× developer buffer (Bio-Rad) at 37 °C for 24 h. After incubation, the gels were stained with Coomassie Brilliant Blue R-250 (Sigma) in a fixing solution containing 10% acetic acid/40% methanol in distilled water. The gels were destained with the fixing solution for 3 h at room temperature. MMP activity was determined as the density of the unstained band.

### 2.13. Statistical Analyses

Statistical significance was evaluated using Student’s *t*-test at a significance level of *p* < 0.05, and the data are expressed as the mean ± S.E. The data were analyzed using ANOVA. Specific contrast analysis was performed using the LSD post hoc test. The survival curves were generated using the Kaplan–Meier method and compared using the log-rank test. All statistical analyses were performed using SAS software (ver. 9.1; SAS institute, Cary, NC, USA).

## 3. Results

### 3.1. Hepatic Function and PRL-1 Expression in Damaged Human Livers

Prior to investigating the function of PRL-1 in liver, we confirmed the parameters of liver function by assessing the levels of total bilirubin (TB), prothrombin (PT), and platelet counts (PLT). TB and PT values were especially increased by cirrhosis and substantially increased in a severe cirrhotic liver (stage 4) compared with stage 0 (* *p* < 0.05). In addition, the PLT value was significantly decreased in stage4 cirrhotic livers compared with a stage 0 liver (* *p* < 0.05, Table 1). As shown in Figure 1A, the expression of PRL-1 mRNAand protein was significantly reduced in stage 4 compared with stage 0 (* *p* < 0.05, Figure 1A,B). Localization of PRL-1 was observed in the cytoplasm and nucleus of hepatocytes in stage 0 liver tissues; however, it was weakly observed in the cytoplasm of hepatocytes in severe cirrhotic liver (stage 4; Figure 1C). These results indicate that decreased PRL-1 expression was involved in liver injury.

### 3.2. PRL-1 Expression Was Decreased in Damaged Rat Livers

To confirm the clear correlation between PRL-1 and liver injury, we generated two rat liver injury models, both carbon tetrachloride (CCl_4_, mild cirrhosis) and BDL (severe cirrhosis). As shown in Figure 1D, collagen accumulation was dramatically increased in the liver tissues of rats in both models compared to the controls. As expected, collagen accumulation was higher in the BDL group than the CCl_4_ group.

Interestingly, the protein expression level of PRL-1 was significantly decreased in both cirrhotic models compared to the control (Figure 1E; * *p* < 0.05). Additionally, its expression slightly decreased in the BDL group compared with the CCl4 group. Localization of PRL-1 was not observed in cirrhotic liver tissues, whereas it was weakly localized in the cytoplasm of hepatocytes in control liver tissues (Figure 1F). These results suggest that PRL-1 could be predominantly expressed in normal hepatocytes and may act as signal transducer during hepatic regeneration after liver damage.

### 3.3. CP-MSC Transplantation Increased the Survival Rate in a BDL Rat Model via Enhanced Hepatic Regeneration

To confirm the therapeutic effect of CP-MSC transplantation in the BDL model, we generated a cirrhotic animal model, as shown in Figure 2A. BDL is the most common model used to induce cholestasis injury in rodents, causing proliferation of biliary epithelial cells and oval cells and resulting in inflammation, hepatocyte apoptosis, and fibrosis [32]. Ten days after BDL, we divided the animals into three groups: the non-transplantation group (BDL); the CP-MSC transplantation group (CP-MSCs); and the WI-38 cell transplantation group (WI-38, negative control). The CP-MSCs and WI-38 cells (2 × 10^6^ cells) were transplanted into the animals, which were sacrificed at 1, 2, 3, and 5 weeks after transplantation. The survival rates of the BDL and WI-38 groups were decreased to zero (*p* < 0.05) at 5 weeks. However, the survival rate in the CP-MSC transplantation group was significantly higher than those of the BDL and WI-38 groups at 5 weeks and was prolonged to 9 weeks with 52% survival rates (Figure 2B). Additionally, the ratio of liver to body weight increased in the BDL, CP-MSC, and WI-38 groups compared to the control group (*p* < 0.05), and no differences among these three experimental groups were observed until 4 weeks. However, the ratio of liver to body weight in the CP-MSC group showed a slightly reduced pattern at 5 weeks after transplantation (Figure 2C). To analyze liver function, we performed an indocyanine green (ICG) clearance assay. ICG clearance was significantly increased in the BDL and cell transplantation groups compared to the control group at all times (*p* < 0.05). However, the value of ICG clearance in the CP-MSC group decreased in a time-dependent manner and was significantly lower than in the BDL and WI-38 groups at 5 weeks after transplantation (*p* < 0.05, Figure 2D).

In addition, we assessed the progression of liver fibrosis in each group using Masson’s trichrome staining. As shown in Figure 2F, the degree of liver fibrosis in the BDL groups with or without transplantation was increased at all times (*p* < 0.05), whereas the degree of fibrosis was significantly decreased in the CP-MSC transplantation group compared to those of the BDL and WI-38 groups until 2 weeks (*p* < 0.05). Furthermore, liver fibrosis had progressed further in the WI-38 transplantation group than the BDL group after 2 weeks (*p* < 0.05; Figure 2E). Next, we investigated whether the efficacy of engraftment affects the survival rate in addition to the liver weight. The number of engrafted CP-MSCs in rat liver was higher than that of WI-38 cells at all times after transplantation (over 2-fold at 1 week), although the number of engrafted CP-MSCs decreased in a time-dependent manner (Figure 3A). These results indicate that CP-MSCs effectively engrafted into an injured rat liver can facilitate the resolution of fibrosis, promoting liver regeneration in the BDL model.

### 3.4. CP-MSC Transplantation Promoted PRL-1 Expression as Well as That of the Migration-Related Rho Family Proteins

Generally, several types of cytokine response are activated during liver injury [33]. Among them, stromal cell-derived factor-1 (SDF-1) binds to CXCR4, which plays an important role in stem cell migration. To investigate whether cytokine induction during liver injury regulates PRL-1 expression in the CP-MSC-transplanted BDL model, we quantified the expression levels of CXCR4 and PRL-1 using Western blotting and performed a correlation analysis. As shown in Appendix A, PRL-1 and CXCR4 levels were decreased simultaneously in the BDL model until 5 weeks (R^2^ = 0.956, *p* = 0.045). Furthermore, the correlation between CXCR4 and PRL-1 expression in the CP-MSC transplantation group was higher than that in the WI-38 group (R^2^ = 0.975, *p* = 0.025) (Appendix A).

Additionally, we confirmed the engraftment of CP-MSCs by PKH 67 labeling. The number of engrafted CP-MSCs in rat liver was higher than that of WI-38 cells at all times after transplantation (over 2-fold at 1 week), although the number of engrafted CP-MSCs decreased in a time-dependent manner (Figure 3A). Additionally, we investigated whether PRL-1 expression affected CP-MSC engraftment via the altenative mRNA and protein expression related to adhesion molecules such as RhoA and ROCK1 in a BDL rat model. The mRNA levels of RhoA and ROCK1 were significantly increased in the transplanted groups with CP-MSCs and WI-38 cells compared to the BDL group (*p* < 0.05, Figure 3C,D). Additionally, their protein levels were significantly increased at 1 and 2 weeks. These findings suggest that CP-MSC transplantation can enhance migration-related signals by increasing cytokine expression, and increasing PRL-1 expression can effectively facilitate the engraftment of CP-MSCs in a BDL rat model via active cross-talk between migration-related Rho family proteins.

### 3.5. Increased PRL-1 Expression by CP-MSCs Is Involved in Regeneration and Proliferation in BDL Rat Liver

Recently, it has been reported that PRL-1 expression is associated with proliferation of liver cells as well as the early events in liver regeneration [34,35]. Therefore, we investigated whether the changes in PRL-1 expression affected the regeneration of damaged hepatocytes in the BDL rat model after CP-MSC transplantation. qRT-PCR analysis revealed that CP-MSC transplantation dramatically increased albumin expression in the BDL rat model compared with the BDL and WI-38 groups at all weeks, although no difference in the protein levels of albumin was found in the CP-MSC groups (* *p* < 0.05, Figure 4A). However, PRL-1 mRNA and protein expression levels in the CP-MSCtransplantation group were significantly increased until 2 weeks compared with the other groups (* *p* < 0.05, Figure 4B).

To analyze the effect of CP-MSCs on the proliferation of damaged hepatocytes, we assessed 5′-bromo-2′-deoxyuridine (BrdU)-incorporated cells in liver tissues by FACS analysis after subcutaneous injection of BrdU. Although BrdU-positive hepatocytes gradually decreased in all groups, including BDL, CP-MSCs, and WI-38, the number of BrdU-positive hepatocytes in the CP-MSC group was higher than that in the BDL and WI-38 groups (Figure 4C). In addition, the number of Ki-67-positive hepatocytes was significantly increased in the CP-MSC transplantation group versus the BDL (* *p* < 0.05) and WI-38 (# *p* < 0.05) groups (Figure 4D).

### 3.6. Down-Regulated PRL-1 Decreased the Migration of CP-MSCs via Migration-Related Rho Family and Matrix Metalloproteinase Protein Expression

To analyze the effect of PRL-1 on the migration ability of CP-MSCs, we performed a migration assay of CP-MSCs using the transwell insert system. Interestingly, the number of invading CP-MSCs was significantly higher than that of WI-38 cells regardless of LCA and siRNA-PRL-1 treatment (Figure 5A,B). The number of invading CP-MSCs was significantly decreased when co-cultured with LCA-treated primary hepatocytes compared with normal primary hepatocytes (*p* < 0.05); otherwise, no difference in invading WI-38 cells was noted because their numbers were substantially lower than those of CP-MSCs. Furthermore, the number of invading CP-MSCs was significantly decreased by treatment with siRNA-PRL-1 compared with no treatment regardless of LCA treatment (* *p* < 0.05). Next, we confirmed the matrix metallopeptidase (MMP) activity by zymography and enzyme-linked immunosorbent assay (ELISA) to analyze the effect of PRL-1 on MMP activity. Our previous studies showed that MMP is a main regulator during migration, and its activities can promote the migration ability of CP-MSCs by interacting with microenvironments via the Rho family [36]. In the ELISA, MMP-2 and MMP-9 levels in CP-MSCs were significantly decreased by LCA treatment compared with no treatment regardless of siRNA-PRL treatment (Figure 5C). MMP-2 expression, but not MMP-9 expression, was significantly decreased by the combined LCA and siRNA-PRL-1 treatment (# *p* < 0.05, Figure 5C). Similar to the ELISA results, MMP-2 activity was significantly decreased by siRNA-PRL-1 treatment compared with no treatment (* *p* < 0.05). Furthermore, MMP-2 activity was significantly decreased by LCA treatment compared with the no treatment group regardless of siRNA-PRL-1 treatment (# *p* < 0.05). However, the MMP activities in WI-38 cells were not significant in any group (Figure 5C). Furthermore, we performed qRT-PCR analysis in CP-MSCs and WI-38 cells that invaded from the upper to the lower chamber. PRL-1 and RhoA mRNA levels were significantly decreased in invading CP-MSCs and WI-38 cells by treatment with siRNA-PRL-1 compared with the no treatment group (* *p* < 0.05). Moreover, PRL-1 and RhoA mRNA levels in CP-MSCs and WI-38 cells treated with LCA were significantly higher compared with no treatment (# *p* < 0.05, Figure 5D,E). ROCK1 expression was decreased in CP-MSCs and WI-38 cells by siRNA-PRL1 treatment compared with the no treatment group. In addition, ROCK1 expression was significantly decreased in invading CP-MSCs by treatment with LCA and a combination with siRNA-PRL-1 and LCA compared with the no treatment group (# *p* < 0.05, Figure 5F). These results suggest that down-regulated PRL-1 expression suppresses the migration of CP-MSCs into damaged hepatocytes via decreased levels of Rho-family-related factors, such as RhoA and ROCK1.

### 3.7. Increased PRL-1 Expression by CP-MSCs Is Involved in the Regeneration and Proliferation of Damaged Hepatocytes

To confirm the effect of PRL-1 on hepatic regeneration, we cultured isolated primary rat hepatocytes with PKH67-labeled CP-MSCs or WI-38 cells after treatment with LCA, and PKH67-labeled CP-MSCs and WI-38 cells were separated using FACS after direct in vitro co-culture (Figure 6A). Albumin and PRL-1 expression levels in isolated hepatocytes treated with siRNA-PRL-1 were decreased compared with a sham group regardless of LCA treatment or co-culture with CP-MSCs (* *p* < 0.05, Figure 6A middle and right panel). However, co-culture with CP-MSCs substantially increased albumin and PRL-1 expression, regardless of siRNA-PRL-1 treatment, compared with LCA treatment (# *p* < 0.05, Figure 6A middle and right). Additionally, we found that albumin and BrdU-positive hepatocytes were significantly increased by indirect co-culture with CP-MSCs compared with LCA-treated hepatocytes (# *p* < 0.05) and WI-38 co-culture († *p* < 0.05; Figure 6B,C). These results suggest that CP-MSCs promote liver regeneration via the up-regulation of hepatocyte proliferation through increased PRL-1 expression in vivo and in vitro.

## 4. Discussion

MSCs have promising therapeutic applications in tissue regeneration because of their abilities of self-renewal, multipotent differentiation into several lineages, and migration to sites of inflammation and injury [37,38]. The therapeutic efficacy of MSCs in repairing damaged tissues depends on strategies that enhance the effects of engraftment via migration of MSCs into target tissues. MSC migration is regulated by multiple processes, responding to secrete various cytokines and growth factors from ischemic and injured tissues [39]. An in vitro study by Ponte et al. showed that 16 growth factors and chemokines affect the migration ability of bone-marrow-derived MSCs (BM-MSCs) [40]. Actually, BM-MSCs are known to be capable of homing to a variety of injured sites in the body, such as myocardial infarction [41], traumatic brain and lung injury, and bone fractures, as shown inin vivostudies. However, knowledge of their molecular process and the mechanism of how MSCs migrate and are guided to the regulator location is still incomplete [42,43,44].

Our previous reports showed that overexpressed PRL-1 expression in CP-MSCs promotes hepatic regeneration via microRNA expression and mitochondrial dynamics in a cirrhotic rat model [45,46]; it could not be decided whether the mechanisms of hepatic regeneration and MSC homing had an endogenous or exogenous effect on engrafted MSCs. Therefore, this study highlights the need to investigate the mechanisms of MSCs homing dependent on PRL-1 expression in the cirrhotic liver model, using distinct separation methods, by sorting the MSCs and hepatic cells. Recently, some researchers have demonstrated that the PRL family members PRL-1, 2, and 3 play a significant role in the development and metastasis of various cancer types [47] and enhance cell migration into endothelial cells in vitro [48]. Luo et al. showed that PRL-1 overexpression was associated with elevated levels of MMP-2 and MMP-9 in HEK293 cells through the activation of p130 Cas and focal adhesion kinase (FAK) [19]. From this evidence, we could infer that PRL-1 expression is involved with the migration processes and mechanism of MSCs.

Based on these findings, we hypothesized that CP-MSC homing depends on PRL-1 expression, which plays a role in liver regeneration directly or indirectly in a damaged liver. In this study, to further elucidate the roles of PRL-1 in MSC migration, we make a comparative study by analyzing the efficacy of CP-MSC engraftment compared with WI-38 cells in a BDL rat model [36]. The numbers of engrafted CP-MSCs were significantly increased on the side of the vessel in the liver tissues of the BDL model, although the activity of engrafted CP-MSCs was decreased in a time-dependent manner. Interestingly, analysis of the scatter plot (Appendix A) revealed that the pattern of PRL-1 expression was correlated with the expression of CXCR4 when CP-MSCs were engrafted into a BDL liver. Therefore, we investigated the expression of focal adhesion molecules and Rho family GTPases mediated by PRL-1 signaling in the liver tissues of a BDL model after CP-MSC transplantation. It has been reported that several PRLs can control cell migration by modulating the phosphorylation of focal adhesion molecules and the cell–extracellular matrix interaction, which can affect actin cytoskeletal changes by regulating the Rho family [49,50,51]. First, we confirmed that PRL-1 induces liver regeneration by regulating MSC homing through RhoA-mediated ROCK1 signaling. In addition, the inhibition of PRL-1 by siRNA-PRL-1 reduced CP-MSC migration into damaged primary rat hepatocytes by decreasing RhoA and ROCK1 levels, as well as reducing MMP-2 and MMP-9 levels. These findings indicate that the expression of the Rho family and MMPs are regulated by PRL-1, which could be a key regulator for chemo-attractive CP-MSC migration in the BDL rat model.

Despite the importance of PRLs in hepatic regeneration, a PRL-mediated mechanism of MSC-based therapeutic intervention is unclear. Therefore, PRLs may be related to the stage of liver cirrhosis as well as hepatic regeneration by CP-MSC transplantation. In this study, we first analyzed the expression of PRL-1 with respect to the stage of cirrhosis in human liver and a BDL rat model. PRL-1 expression was decreased in liver tissues of humans and rats, depending on the stage of cirrhosis. Recently, Dumaualand their colleagues reported that PRL-1 was weakly expressed in the cytoplasm and nucleus of normal liver hepatocytes, and in less than 30% of hepatocytes, it was strongly expressed in the nucleus [52]. According to previous studies, the levels and the localization of PRL-1 expression involved in liver regeneration are still controversial because there are difference between disease models, disease states, and species. Hence, further studies on their expression and regulation mechanismare needed. Second, we investigated the effect of PRL-1 on hepatic regeneration in damaged liver tissues in a BDL model transplanted with CP-MSCs and compared it to the BDL alone and WI-38-transplanted groups. The expression levels of albumin and PRL-1 were simultaneously increased, and proliferation of hepatocytes was increased by CP-MSC transplantation. In contrast, down-regulation of PRL-1 by siRNA-PRL1 induced decreased proliferation of hepatocytes as well as the expression of albumin regardless of CP-MSC co-culture. PRL-1 expression was decreased in hepatocytes in a time-dependent manner after CP-MSC transplantation in vivo. PRL-1 was originally identified as an early gene because it correlated with immediate-early transcription factors through the cell cycle during liver regeneration [53,54,55]. Additionally, the mitochondrial biogenesis effects of PRL-1 have been reported recently, suggesting the therapeutic effect of PRL-1 and emphasizing the clinical impact [56,57]. These results suggest that up-regulation of PRL-1 was induced immediately for liver regeneration after CP-MSC transplantation. Our in vitro data show that the proliferation of primary hepatocytes co-cultured with CP-MSCs was significantly increased regardless of LCA or siRNA-PRL-1 treatment.

In conclusion, our data establish the effects of CP-MSC transplantation on liver regeneration, as outlined in the schematic in Figure 7. Migration of CP-MSCs was regulated by PRL-1-mediated small GTPase molecules (e.g., RhoA, ROCK1) and MMP-2 activity in the liver tissues of a BDL rat model. Engrafted CP-MSCs increased PRL-1 expression and proliferation of hepatocytes. In addition, increased PRL-1 expression enhanced the regeneration of a damaged liver by increasing albumin in hepatocytes. These findings are the first report to reveal the dual function of PRL-1 in the migration and therapeutic efficacy of CP-MSCs in damaged liver tissues. Therefore, these results elucidate a fundamental mechanism for the therapeutic effects of PRL-1 in hepatic diseases by CP-MSC transplantation, and support the development of cell-based therapeutic strategies for regenerative medicine in liver disease using CP-MSCs.

## Figures and Tables

**Figure 1 cells-10-02530-f001:**
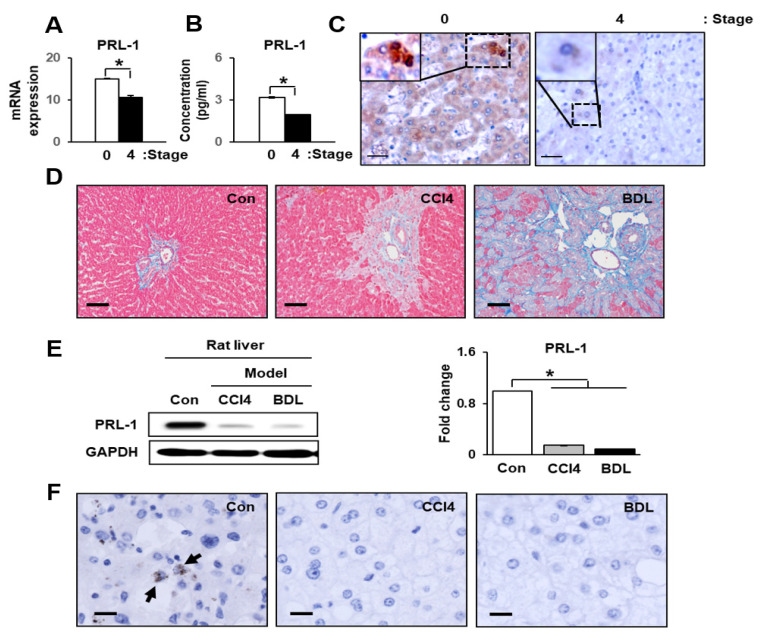
PRL-1 correlates with liver injury in a human and animal disease model. (**A**) The relative expression and (**B**) concentration of PRL-1 in fibrotic human liver (e.g., stages 4) compared with stage 0 human liver by quantitative RT-PCR and ELISA, respectively. (**C**) Localization and expression of PRL-1 were detected in non-fibrotic human liver (*n* = 5; stage 0) and severe cirrhosis (*n* = 5; stage 4). Liver sections were stained using anti-PRL-1. PRL-1-positive signals (DAB chromogen in brown) were detected in the cytoplasm and nucleus of hepatocytes in stage 0 human liver. The scale bar represents 75 µm. (**D**) The degree of liver fibrosis was analyzed by Masson’s trichrome staining (200×). PRL-1 expression was evaluated by quantitative (**E**) immunoblotting and (**F**) immunohistochemistry. GAPDH was used as an internal control. Liver sections were stained using anti-PRL-1. A hematoxylin counterstain was used. The scale bar represents 75 µm. Filled arrows indicate the PRL-1-positive hepatocytes in rat liver tissue. Each experiment was repeated three times. The data are expressed as the mean ± SEM (* *p* < 0.05).

**Figure 2 cells-10-02530-f002:**
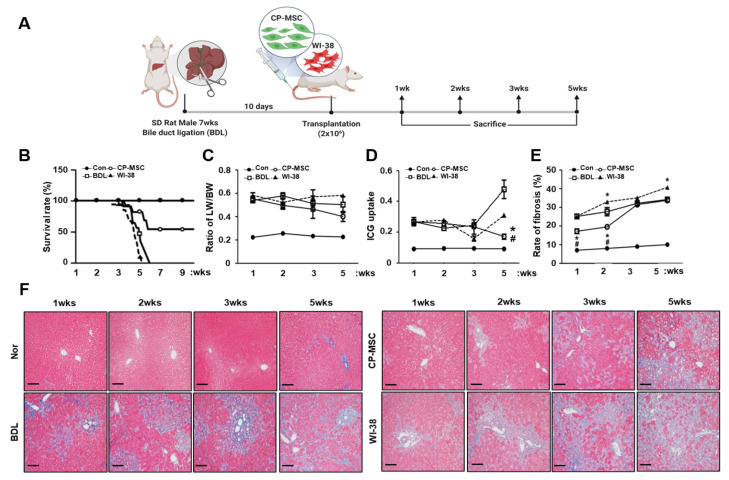
Engrafted CP-MSCs promote liver function in the liver injury rat model with BDL. (**A**) A schematic diagram describing the rat model of BDL. After BDL, the cells (CP-MSCs and WI-38; 2 × 10^6^) were transplanted into the rats via the tail vein, and the animals were sacrificed at 1, 2, 3, and 5 weeks. (**B**) The survival curve of rats in 4 groups (Con, BDL, CP-MSCs and WI-38 group) over 10 weeks. The rat survival rate was monitored daily and is expressed by Kaplan–Meier survival curves. (**C**) Liver/body weight ratios were determined, and (**D**) an ICG clearance test was performed in the control group (Con), the BDL group (BDL), the CP-MSC transplantation group (CP-MSCs), and the WI-38 transplantation group (WI-38) at 1, 2, 3, and 5 weeks after cell transplantation. (**E**) The ratio of fibrotic/normal hepatocytes was analyzed by the fibrosis scoring system using ImageJ software through (**F**) Masson’s trichrome staining. Each experiment was repeated at least three times. The data are expressed as the mean ± SEM (* *p* < 0.05 compared with the BDL group; # *p* < 0.05 compared with the WI-38transplantation group). The scale bar represents 75 µm.

**Figure 3 cells-10-02530-f003:**
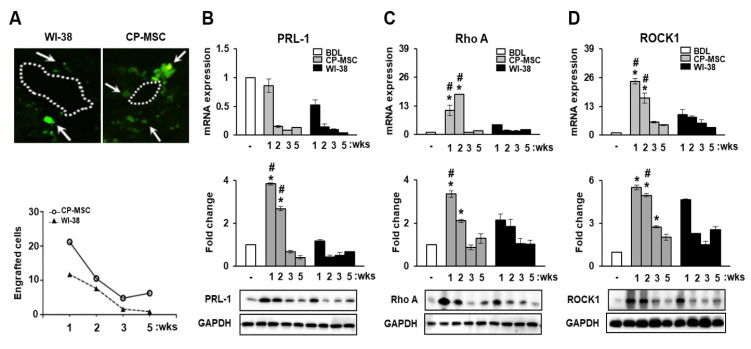
PRL-1-mediated signal induces the homing of CP-MSCs into the BDL-injured rat liver via Rho family proteins. (**A**) PKH67-labeled CP-MSCs and WI-38 cells engrafted into BDL-injured rat liver tissue by immunofluorescence at 5weeks (upper) and FACS (lower). The arrow indicates the engrafted cells near a liver vein. The scale bar represents 50 µm. Expression levels of (**B**) PRL-1, (**C**) Rho A, and (**D**) ROCK1 were determined by quantitative RT-PCR (upper) and Western blotting (lower) in BDL (1 weeks), CP-MSCs, and WI-38-transplanted livers (1, 2, 3, and 5 weeks). Total RNA and protein from liver in each group were pooled (at least five rats per group). GAPDH was used as an internal control. Each experiment was repeated three times. The data are expressed as the mean ± SEM (* *p* < 0.05 compared with the BDL group; # *p* < 0.05 compared with the WI-38transplantation group).

**Figure 4 cells-10-02530-f004:**
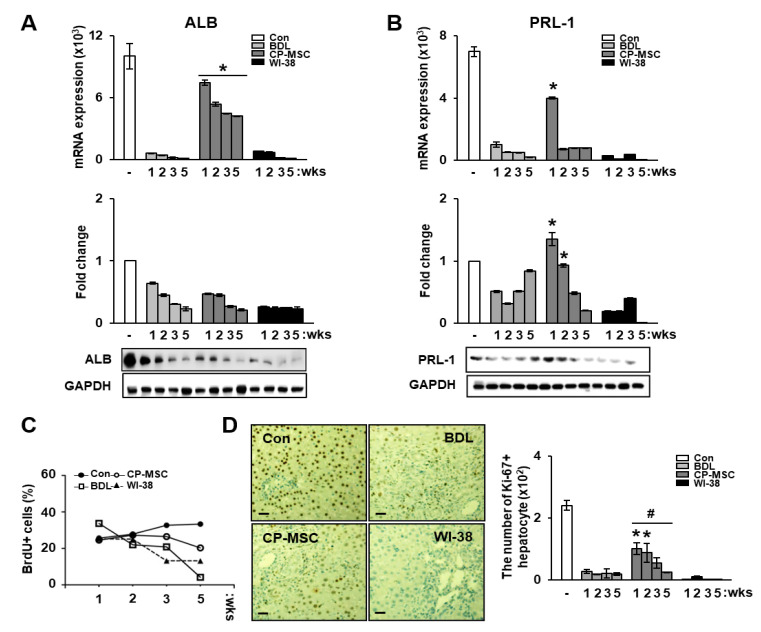
Effect of CP-MSCs on the proliferation of damaged rat hepatocytes in vivo and in vitro. mRNA and protein levels of (**A**) albumin (ALB) and (**B**) PRL-1 in hepatocyte-only extracts isolated from a BDL-injured rat model after cell transplantation were analyzed by qRT-PCR (upper) and Western blotting (lower). Equal amounts of RNA and protein were pooled in each group. GAPDH was used as an internal control. (**C**) BrdU-positive hepatocyte cells isolated from BrdU-injected BDL rats were gated and quantified using mouse anti-BrdU in normal, BDL, CP-MSC transplantation, and WI-38 transplantation groups at 1, 2, 3, and 5 weeks after cell transplantation. (**D**) Localization and quantification of Ki-67-positive hepatocytes in the BDL model after transplantation of CP-MSCs and WI-38 cells. Liver sections were stained using anti-rat Ki-67; Ki-67-positive signals (DAB chromogen in brown) were detected and counted using Image J software. The scale bar represents 75 µm. Each experiment was repeated at least three times. The data are expressed as the mean ± SEM (* *p* < 0.05 compared with the BDL group; # *p* < 0.05 compared with the WI-38 transplantation group).

**Figure 5 cells-10-02530-f005:**
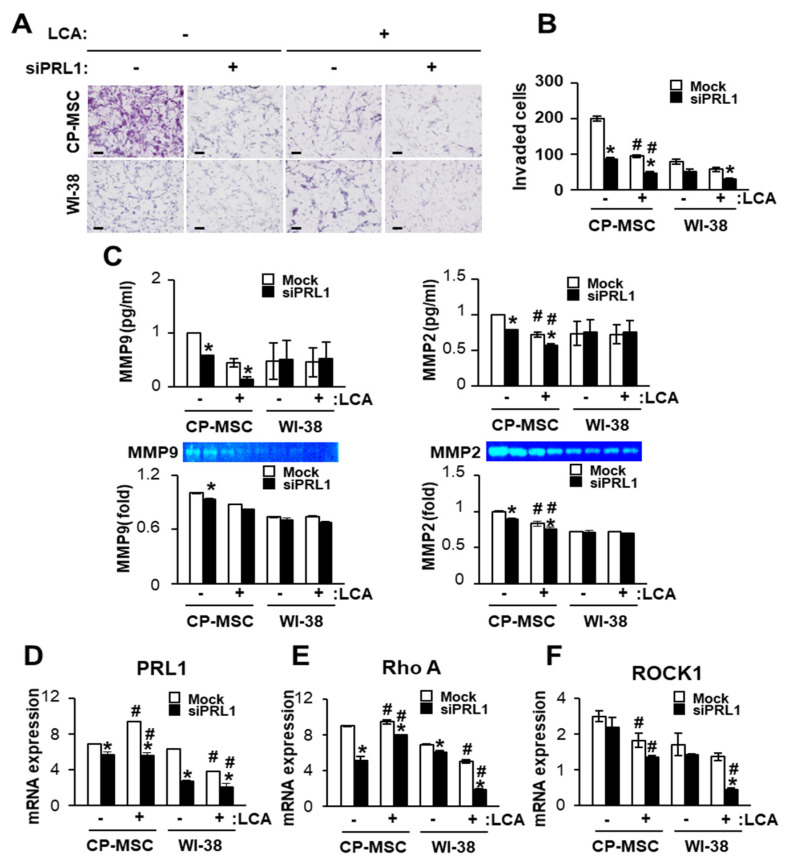
PRL-1 regulates CP-MSC migration into damaged rat hepatocytes through MMPs and the Rho family in vitro system. (**A**) Images and (**B**) quantification of invading CP-MSCs or WI-38 cells in the transwell insert system co-cultured with rat primary hepatocytes treated or untreated with siRNA-PRL-1 after 12 h of treatment with or without LCA. The invading cells were counted. The scale bar represents 100 µm. (**C**) Enzyme activities of MMP-9 and MMP-2 in conditioned media collected from the transwell insert system treated with or without siRNA-PRL-1 after 12 h of treatment with or without LCA by ELISA (upper channel) and zymography (down channel). Each bar represents the fold changes in MMP-9 and -2 expression compared with the siRNA-PRL-1 and LCA non-treated group. mRNA levels of (**D**) PRL-1, (**E**) Rho A, and (**F**) ROCK1 in invading CP-MSCs or WI-38 cells from the upper chamber to the lower chamber after treatment with or without siRNA-PRL-1 toward rat primary hepatocytes treated with or without LCA. Total RNA from the invading cells in each group was pooled. Each experiment was repeated at least three times. The data are expressed as the mean ± SEM (* *p* < 0.05 compared with siRNA-PRL-1treatment group; # *p* < 0.05 compared with the LCA treatment group). siPRL1, siRNA-PRL1; LCA, lithocholic acid.

**Figure 6 cells-10-02530-f006:**
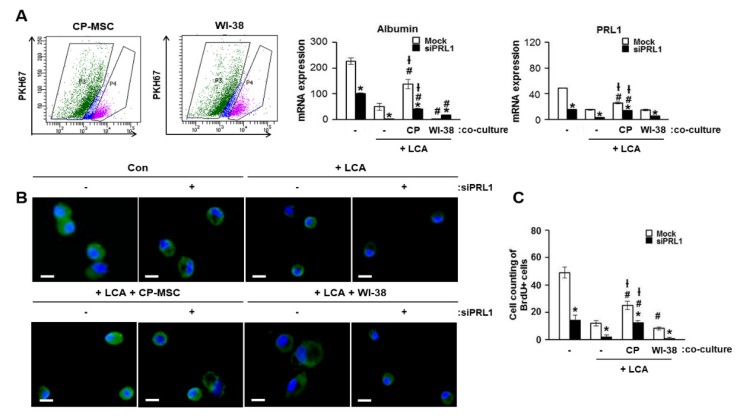
Increased PRL-1 expression by CP-MSCs correlates with hepatocyte proliferation. (**A**) The fraction of PKH67-negative primary rat hepatocytes was sorted by flow cytometry in the CP-MSC and WI-38 direct co-cultured group 24 h after treatment with siRNA-PRL-1. mRNA expression of albumin and PRL-1 in sorted primary rat hepatocytes treated with siRNA-PRL-1 after direct co-culture with CP-MSCs and WI-38 was examined by qRT-PCR. The localization of (**B**) albumin and (**C**) BrdU-positive hepatocytes was determined with immunofluorescence in rat primary hepatocytes isolated from a BDL rat model treated or untreated with siRNA-PRL-1 after culturing with or without CP-MSCs or WI-38 cells. Rat primary hepatocytes expressed green fluorescent protein and were stained for ALB (green) with 4ʹ,6-diamidino-2-phenylinodole (DAPI) (blue). A representative overlay image is shown. The scale bar represents 50 µm. The data are expressed as the mean ± SEM (* *p* < 0.05 compared with siRNA-PRL-1treatment group; # *p* < 0.05 compared with the LCA treatment group; † *p* < 0.05 compared with the treatment regardless of LCA or siRNA-PRL-1). CP, CP-MSCs; siPRL1, siRNA-PRL-1; LCA, lithocholic acid.

**Figure 7 cells-10-02530-f007:**
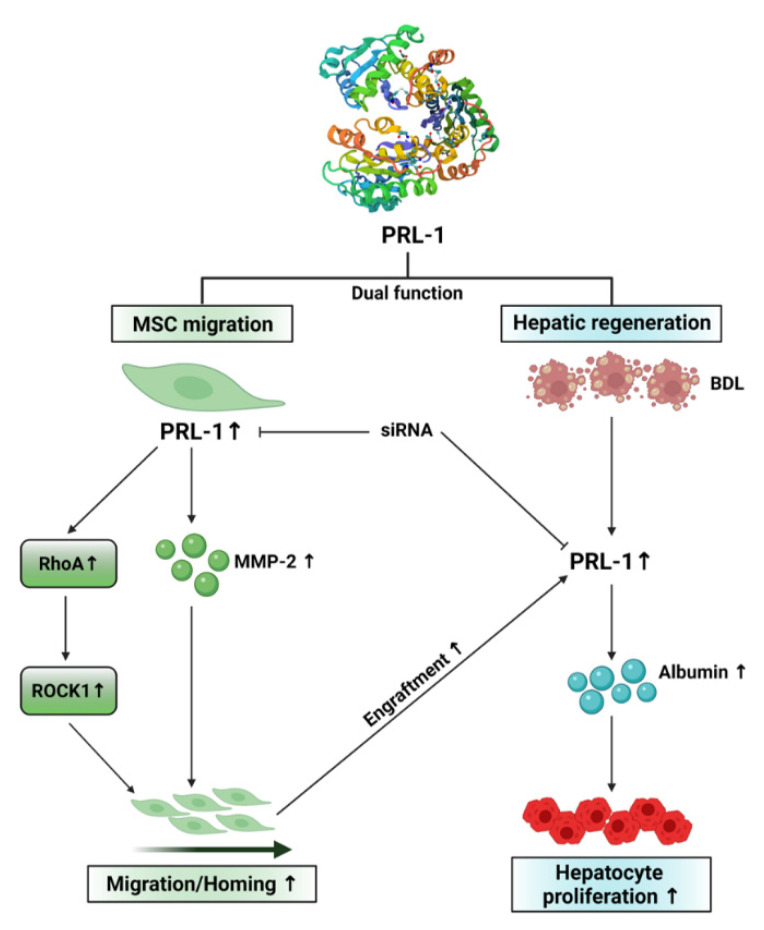
A proposed model of CP-MSC-induced hepatic regeneration through regulation of PRL-1 expression. Migration of transplanted CP-MSCs to the damaged liver tissues was regulated by a PRL-1-mediated Rho family signaling pathway. Engrafted CP-MSCs stimulated hepatic regeneration in damaged liver tissues through increased PRL-1 expression and enhanced hepatic proliferation.

**Table 1 cells-10-02530-t001:** Liver function test in fibrotic human livers.

Parameter	Stage				
	0 (*n* = 5)	1 (*n* = 5)	2 (*n* = 5)	3 (*n* = 5)	4 (*n* = 5)
AST	162.8 ± 78.9	78.4 ± 9.6	97.4 ± 21.5	85.6 ± 19.8	71.4 ± 11.4
ALT	104.9 ± 43.4	94 ± 28.7	42.8 ± 6.9	52 ± 15.6	68.6 ± 24.0
TB	0.68 ± 0.2	0.86 ± 0.2	2.08 ± 0.8	2.42 ± 1.0	2.78 ± 0.9 ^a^
γGT	330 ± 110	133 ± 70.2	383 ± 228	744 ± 335	346 ± 231
PT	0.92 ± 0.04	0.92 ± 0.02	1.04 ± 0.04	0.93 ± 0.03	1.31 ± 0.07 ^b^
PLT	349,400 ± 98,522	349,800 ± 55,757	181,400 ± 55,275	160,200 ± 40,786	98,000 ± 14,638 ^c^

Data are shown as the mean (range). AST, aspartate aminotransferase; ALT, alanine aminotransferase; TB, total bilirubin; γGT, gamma glutamyl transpeptidase; PT, prothrombin time; PLT, platelet count; *n*, number of patients. ^a^
*p* < 0.05 compared to 0 stage; ^b^
*p* < 0.05 compared to 0, 1, and 3 stage; ^c^
*p* < 0.05 compared to 0, 1, 2, and 3 stage.

## Data Availability

Not applicable.

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
