# Peer review of "Increased Phosphatase of Regenerating Liver-1 by Placental Stem Cells Promotes Hepatic Regeneration in a Bile-Duct-Ligated Rat Model"

_cells, 2021, doi:10.3390/cells10102530_

Round 1

Reviewer 1 Report

The quality of the manuscript has been improved and is now suitable for publication.

Author Response

Manuscript ID: Cells-1359822

Author's Reply to the Review Report (Reviewer 1)

The quality of the manuscript has been improved and is now suitable for publication.

Author’s reply:
We appreciate your positive comment.

Reviewer 2 Report

Review: Increased Phosphatase of Regenerating Liver-1 by Placental 2 Stem Cells Promotes Hepatic Regeneration in a Bile 3 Duct-Ligated Rat Model by Choi et al

The authors present a role for the phosphatase of regenerating liver-1 (PRL-1) protein in the recovery rats with bile duct ligation surgeries performed to induce liver damage. The mechanisms and roles of various proteins and signaling pathways associated with liver regeneration, damage repair, and inflammation remain only partially understood, so a paper helping to fill this knowledge gap is valuable. However, the authors did not present a clear enough role for PRL-1 to convince the reviewer that this pathway is important. For this reason, and the major concerns below, I do not find this article acceptable for publication.

Major concerns:

                  - Many, if not all, of the associations between PRL-1 and other proteins of interest (CXCR4, RhoA, ROCK1, Albumin) are inferred based upon increases in abundance after a major traumatic surgery. It is likely that many liver proteins will increase or decrease in abundance after BDL, but this does not mean they are associated with the function of PRL-1 or vice versa.  In order to support a causal role, evidence must be direct, not correlative.

                  - The authors seem to equate stem cell engraftment following liver damage to regeneration. The only measures for liver regeneration mentioned are the markers Ki-67 and BRDU, both of which saw lower amounts of staining in surgery treated animals than the resting control, which should be post-replicative. Statements regarding liver regeneration are not found in evidence in this article.  Significantly, it is well established that the regenerative response in liver is predominated by mature differentiated cell types, not by “stem” or progenitor cells. Moreover, I am aware of no currently accepted (i.e., not subsequently refuted in peer-reviewed publications) showing a role for exogenous (i.e., not liver-resident) stem/progenitor cells in liver regeneration.

                  - Many of the observations in gene and protein expression after stem cell engraftment (CP-MSC’s) are also noted in the negative control (WI-38) cells. Similarly, many of the PRL-1 specific functions change regardless of the delivery of mock or siRNA-PRL-1, indicating they are PRL-1 independent functions.

                  - The rationale for measuring the association between PRL-1 and proteins like the MMP’s is not provided in the Introduction or Results sections, and only briefly expressed in the Discussion. Furthermore, the results section has very little explanation of what the results mean, except for a single sentence at the end of each paragraph. For the most part, conclusions are inferred rather than presented. It was difficult to follow the authors rationale for experiments or what each set of data meant relative either supporting or rejecting to their hypotheses.

                  - Much of the data seems to be contradictory. For instance, the observation on line 225 that PRL-1 is only weakly localized to the cytosol in the injured liver in Fig 1C, but on line 497, the authors state that PRL-1 localizes to the nucleus upon injury. Similarly, data and trends seem to change based on factors not controlled for, like the increased migration of the experimental CP-MSC’s vs WI-38 controls. Similarly, it is unclear at what points hepatic PRL-1 is being measured vs PRL-1 in CP-MSC’s or WI-38.

Minor concerns:

                  - English language and grammar need revision throughout the manuscript

                  - The sentence on line 286 appears again on line 305

- On line 278, the authors measure ICG clearance in WI-38 and control groups at week 5, however on line 267, the authors note that survival rates for both groups were at 0 by week 5

- Line 238 – the sentence linking PRL-1 to signal transduction is not presented with supporting data. The authors have interpreted IHC localization to infer a signaling pathway during regeneration without any data

- Error bars for all WI-38 experiments in Fig 5 are significantly larger than all CP-MSC experiments, even for mock treated cells without LCA. More replicates should be added to decrease the substantial error in these measurements.

- Line 482 – if the authors are ruling out regeneration by FAK, this data needs to be shown

- Line 499 states the authors observed nuclear localization of PRL-1, however this is not shown in any data presented, only the decrease in cytosolic PRL-1

Reviewer 3 Report

Although it is a very complete work experimentally and of great interest, I have some doubts about it, which I detail below.

  • Add transplantation after stem cells, please.
  • Indicate that one of the problems encountered for whole liver transplantation is also the shortage of organs for transplantation, please.
  • Line 73: Homing should be lowercase.
  • Line 82: define BDL
  • The intro does not indicate more than the transplant study, at least that is my understanding, it does not say anywhere what is intended to be determined in human tissue or co-cultures.
  • To my view, the methodology section is disorganized and difficult to understand. In addition, what authors determine and why should be all justified in the intro as well as in the objectives (as far as co-cultures and human histology is concerned)
  • Why are WI-38 cells used as a control?
  • Section 2.3. Immunohistochemical staining: which specimens?
  • Section 2.4. Cultivation of CP-MSCs and primary hepatocytes: are these placentas the same ones from which the cells to be transplanted are extracted?
  • Line 138: Isolated primary hepatocytes were from humans?
  • Why did the authors choose the LSD post hoc test?
  • Section: 3.1. Hepatic function and PRL-1 expression in damaged livers of humans.
  • Was any statistical correlation study of these parameters done?
  • Line 230: How was the degree of cirrhosis quantified?
  • I disagree with the use of a comparison group to assess PRL-1 expression in the BDL model. One is a chemical model and the other is surgical. Maybe modulated pathways are different.

Major concern

In my view, there is a mixture of models in this manuscript. The part in vitro would be part of another manuscript. All the studies (westerns, histology, mRNA expression) as well as serum liver functionality parameters should be performed  on the livers and serum of transplanted and no transplanted . Using as damage control rats with BDL without transplantation.

Round 2

Reviewer 2 Report

Response to Author’s replies

In reviewing the previous submission, I raised fiver major concerns and several minor points. These were very serious concerns that resulted in my decision to recommend rejection of the submission. While the authors have provided lengthy responses and numerous figures in their rebuttal to the the raised concerns, these responses do not address the concerns that I raised. Most responses and data provided in their rebuttal is irrelevant to the serious flaws in the paper that I had commented on previously. Therefore, my initial decision to reject the article remains the same. The paper should be rejected.

Point 1: The authors have not addressed the concern raised in major point 1. mRNA expression of albumin is not significantly different between Naive and PRL-1+ TTx models and expression of ROCK1, RhoA, CDK4 and Cyclin D1 are all increased in the Naive TTx model as well as the TTx PRL-1 overexpression model. This reiterates the major concern that the correlations implied in this paper are merely coincidental, rather than causal.

Point 2: The authors have not addressed the major concern raised in point 2. While the authors may have used Ki-67 and BRDU in previous publications, its use in this publication does not support the hypothesis of liver regeneration.  In short, the authors' rebuttal and references are irrelevant to the criticism raised. Furthermore, the authors did not either acknowledge or address my concern about there being no currently recognized published precedence for non-resident stem cells - placental stem cells (this study) or hematopoietic stem cells in the infamous and discredited studies that came out 2 decades ago - contributing substantially to liver-mass during regeneration. The data in this paper do support this conclusion.  In considering the skepticism that exists in the field, reviewers should demand very careful, exhaustive, thorough, and convincing evidence to support such claims. Such evidence is not found in this paper.

Point 3: The authors have pointed out a piece of data in Fig 4D that they feel refutes the raised concern, however I feel this only solidifies it. It is clear that the observed drop in PRL1 mRNA expression does not change between the experimental CP-MSC and control WI-38 cells. This is repeated in figures 1B, C, D, and 5D, E and F

Point 4: This comment was only partially addressed in one section of the paper. The concern, however, applied to the introduction, results and discussion sections

Point 5: While the authors addressed one part of the comment, the confusion about PRL-1 localization, other parts of the comment were not addressed.

Other points:

- While the English language and grammar are improved in this draft, it is still not acceptable for this publication

- The authors have not clarified how they are measuring ICG clearance in a group of animals that are not alive at 5 weeks when the measurement is taken

- If variations in enzyme activity of MMP2 and MMP9 are to blame for the large error bars of Fig 5, this should be the same in the CP-MSC groups as well

Author Response

Dear Reviewer,

We sincerely apologize for not explaining enough about the Reviewer’s comments in the first revision. Nevertheless, I would like to thank the Reviewer for your sincere advice. Although there is a limited revision periods (10 days) for the issues pointed out by the Reviewers, we tried to do our best to make the response and expect the Reviewer’s positive to consider our responses.   

Please enclosed 2nd revision letter.

Very sincerely yours,

Gi Jin Kim, Ph.D.

Associate Professor

Reviewer 3 Report

Regarding the following statement:

"Author’s reply: We agree with your opinion. However, we confirmed the expression of
PRL-1 in chemical and surgical cirrhotic models, comparing normal rat liver. Its expression in both models was similar. Therefore, we thought that the modulated pathways by PRL-1 could be similar between both models."

In my opinion this point should be included in the discussion since in both models the modulated signaling pathways may be the same, but it is still a hypothesis.

Author Response

Dear Reviewer,

We greatly appreciate your careful evaluation of our manuscript (Cells-1359822) entitled: “Increased Phosphatase of Regenerating Liver-1 by Placental Stem Cells Promotes Hepatic Regeneration in a Bile Duct-Ligated Rat Model.” We were really encouraged by the reviewers’ positive comments and constructive suggestions. I am happy to report that we have successfully addressed all issues and concerns through additional data and subsequent revision of our manuscript, as detailed in the following response page. As Reviewer’s commented, we corrected it clearly stating with each comment and changes are highlighted in red in the revised manuscript. 

Very sincerely yours,

Gi Jin Kim, Ph.D.

Associate Professor

This manuscript is a resubmission of an earlier submission. The following is a list of the peer review reports and author responses from that submission.

Round 1

Reviewer 1 Report

In this paper, Choi and collaborators analyzed the effect of placental stem cells in promoting liver regeneration, indicating PRL-1 as the pharmacological target of this cell therapy. 

The work is interesting. However, I have some concerns that should be solved before publication. 

Experimental issues:

1) The authors did not consider the role of non-parenchymal cells in this process. Several recent studies indicate that HSCs and KC play pivotal roles in liver regeneration. Their role has not been evaluated in this work, and not even discussed based on data present in the literature. This should be added. 

2) Although mentioning the CCl4 model, the authors focus this research on the BDL. They should better explain the reason of their choice. Some efficacy data in another model of liver injury would definitely increase the value of this study.

Methods: some paragraphs are extremely confused (e.g. 2.2). Please check and fix this problem.

Refs: the references are not up to date, only a limited number of the cited papers has been published in the last 2-3 years. Liver regeneration and stem cell therapy is a field which is currently extensively investigated. Please update the literature cited in this paper. 

Author Response

Dear Reviewer,

Thank you for your kind comments.

We were really encouraged by the reviewers’ positive comments and constructive suggestions. I am happy to report that we have successfully addressed all issues and concerns through additional data and subsequent revision of our manuscript, as detailed in the following response page. Changes are highlighted in red in the revised manuscript.

Sincerely yours,

Gi Jin Kim

Reviewer 2 Report

Dear authors, although the presented conclusions of the manuscript are potentially interesting in the field of cancer and the biology of PRLs, there are several important concerns about the information presented.

In general the introduction should be extended to explain about the role of some elements that appear lately in the text (for example metaloproteases). Also in the text we miss the meaning of for example LCA treatment. Referring in the intro the significance of this treatment would help to follow the manuscript, than in its current form is difficult.

Along the text there are some typos that must be corrected (for example ‘anlaysis’ in 2.6 or ‘alterative’ in line 279)

In general methods are not sufficiently detailed, and the statistical analysis in different figures is not enough

Regarding figure 1

As extensively documented in the literature, the size of PRL-1 is around 20 kDa. However, authors show a band of 42 kDa. There are several commercially available antibodies, which can be used rendering a band of around 20kDa. This should be corrected or explain (and show) why PRL-1 is apparently a protein of 40 kD in cells analyzed. Also, the molecular weigh markers in the gels must be shown along the manuscript.

The number of experiments done should be indicated in figure legend

It would be also interesting to test in these systems the expression of PRL-2 and PRL-3. Is is apparently done (as claimed in the discussion), but not shown, why? We find this interesting enough.

There is no protocol for Figure 1C and should be provided. What anti-PRL-1 antibody is used? The same used for WB?

Also in lane 208-209 authors claim that ‘PRL-1 has a role in the maintenance of normal hepatic function’. However, these results show a correlation and only suggest this affirmation. The sentence should be re-formulated.

Regarding Fig 2

Why WI-38 was used? Is it a good control for the author’s assays? This should be explained.

Regarding this claim: ‘However, the ratio of liver to body weight in the CP-MSC group was substantially reduced at 5 weeks after transplantation (Fig. 2C).’ (line 245-246). Are the differences statistically significant? Is the result enough to talk about a substantial reduction? Authors should provide statistics or reconsider this.

Regarding this score ‘The ratio of fibrotic/normal hepatocytes was analyzed by the fibrosis scoring system using ImageJ software (right)’ (line 223-224) Please explain how the score is obtained.

How are quantified the engrafted cells in 2F? What does the dashed line means in pictures? The graph is a representative experiment? The number of experiments should be indicated. Graphs with dispersion/error should be used.

Please explain in the main text the use of PKH67, is just cited in the figure legend.

Supplementary figure 1A

Regarding this sentence: ‘Furthermore, the correlation between CXCR4 and PRL-1 expression in the CP-MSC transplantation group was higher than that 275 in the WI-38 group (R2=0.975, p=0.025) (Supplementary Fig. 1A).’ How many experiments were done? Can be this hypothesis statistically compared. I honestly do not see a great difference between these samples. An example of each condition should be provided.

Where is supplementary figure 1B, if it exist it must be provided or delete the reference form the main text?

Regarding Fig3

Are statistically different BDL and WI-38 comparisons? Please clarify this and if it is necessary correct the text (line 281). Why BDL does not have data along the different weeks?

What adhesion molecule is Nov?, it is not mention in the introduction.

Please RhoA and Rock1 are not adhesion molecules. The text should be corrected (line 279). In the same line when authors say ‘alterative’ I guess they mean alternative. Why alternative? Or authors mean altered. Sorry, I do not understand well the meaning of this sentence.

The initial increment in the expression of the mRNA/protein studied decreased along the weeks. However this result is not described or discussed in the text. This should be done. Also, in Fig3B the increment in PRL-1 protein does not correlate with an increase in the mRNA level in the graph. Again as in fig1 the size of PRL-1 should not be 42 kD. This is a problem in this manuscript. Again, how many experiments were done in this figure. It should be stated in the figure legend.

Data presented in figure 3 do not mean that ‘PRL-1-mediated signal induces the homing of CP-MSCs into the BDL-injured rat liver via Rho family proteins’ as says the title of the figure legend.

Regarding Fig 4

The efficiency of PRL-1 protein down-modulation after siRNA treatment should be shown. This would be very important to show that the protein of 42 kD is actually PRL-1 (To be honest it is difficult to trust). Due to the discrepancy with many already published papers another antibody should be used in addition to the one used in this manuscript. This is important. Again, the number of the experiments performed in panels of figure 4 should be indicated.

Regarding Fig 5B

the same problem of PRL-1 size applies (Even 38 kD is not the size)

Is it possible to show the FACS data in figure 5C?

When Ki-67 is used, explain what are you studying (proliferation)

Regarding Fig 6

In figure 6A what is represented in the x axis of the plot? In the figure legend authors say that there is a siRNA treatment. However, this is not explained in the main text

Again, the down-modulation of the PRL-1 protein after siRNA treatment should be shown. For how long is the siRNA treatment applied before the experiment? Here, in contrast to data shown before there are changes in mRNA levels. This is very important and should be discussed.

What antibody is used in fig6F

Discussion

Authors talk about the nuclear translocation of PRL-1. However, this is not properly studied not quantified in this work. This should be quantified or cited a manuscript where it was previously demonstrated. If this ara data presented in this manuscript it should be done properly.

In our opinion data not shown should be actually included in the manuscript. At least concerning the expression PRLs

It would be nice to have further discussion about the PRL-1-based therapeutic strategies proposed by the authors

Author Response

(The authors gave the same response as above.)

Reviewer 3 Report

This original paper opens a line of research in liver regeneration that is essential in the clinical application of liver resection for tumor pathology. The Materials and Methods and Results sections are adequate. Perhaps under Discussion I would be more emphasis on the clinical impact of this research.

Author Response

(The authors gave the same response as above.)

Round 2

Reviewer 2 Report

Dear Authors,

I am afraid to say that, although the manuscript improved after some corrections the authors did, the manuscript should not be accepted in its current form.

My major concern about the experiments for PRL-1 protein detection and siRNA are still pending. Now authors claim that the protein has 30 kDa. However, this protein is detected as a band even lower than 20kDa. Why don't you sow the molecular weigh markers in your blots? Most importantly, the downmodulation of PRL-1 protein after siRNA treatment must be shown. Otherwise we can not relay the conclusions reached. I asked for this and authors still do not show this important information. Thus, still feel the manuscript should be rejected. Also, methods are still incomplete (for example what is mock in siRNA experiments? did you use an scramble siRNA control?). On the top, other issues I pointed out are not completely solved.